# Medicine authentication technology as a counterfeit medicine-detection tool: a Delphi method study to establish expert opinion on manual medicine authentication technology in secondary care

Bernard Naughton,[1,2,3] Lindsey Roberts,[4] Sue Dopson,[2] David Brindley,[2,5,6,7,8,9] Stephen Chapman[1,5]

For numbered affiliations see end of article.

Correspondence to
Bernard Naughton;
bernard.naughton@sbs.ox.ac.uk

## ABSTRACT

**Objectives:** This study aims to establish expert opinion and potential improvements for the Falsified Medicines Directive mandated medicines authentication technology.

**Design and intervention:** A two-round Delphi method study using an online questionnaire.

**Setting:** Large National Health Service (NHS) foundation trust teaching hospital.

**Participants:** Secondary care pharmacists and accredited checking technicians.

**Primary outcome measures:** Seven-point rating scale answers which reached a consensus of 70–80% with a standard deviation (SD) of <1.0. Likert scale questions which reached a consensus of 70–80%, a SD of <1.0 and classified as important according to study criteria.

**Results:** Consensus expert opinion has described database cross-checking technology as quick and user friendly and suggested the inclusion of an audio signal to further support the detection of counterfeit medicines in secondary care (70% consensus, 0.9 SD); other important consensus with a SD of <1.0 included reviewing the colour and information in warning pop up screens to ensure they were not mistaken for the 'already dispensed here' pop up, encouraging the dispenser/checker to act on the warnings and making it mandatory to complete an 'action taken' documentation process to improve the quarantine of potentially counterfeit, expired or recalled medicines.

**Conclusions:** This paper informs key opinion leaders and decision makers as to the positives and negatives of medicines authentication technology from an operator's perspective and suggests the adjustments which may be required to improve operator compliance and the detection of counterfeit medicines in the secondary care sector.

## Strengths and limitations of this study

- This study is the first of its kind to obtain consensus regarding improvements necessary for medicines authentication technology, which is a Pan-European approach to counterfeit medicine detection; driven by a European directive.
- The key consensus improvements for consideration have narrow SDs of ≤1.0.
- This study focused on a serialisation and authentication technology provided by one of the three European Medicines Verification Organisation 'blueprint providers'.
- At the time of research, authentication technology was available in only one UK hospital; as such, the available participants were limited.
- This study would require implementation on a larger sample once suitable participants became available.

## INTRODUCTION

According to the European Commission, a falsified medicine is a fake medicine that passes itself off as a real authorised medicine;[1] according to the Food and Drug Administration (FDA), a counterfeit medicine is a 'fake medicine' that may be contaminated or contain the wrong or no active ingredient. It could have the right active ingredient but at the wrong dose.[2] According to international intellectual property law, a 'counterfeit' is one that infringes a trademark by bearing an identical or near-identical mark.[3] In contrast, the World Health Organization (WHO) uses the term SSFFC which stands for substandard, spurious, falsely labelled,

falsified and counterfeit medicinal products.[4] The European Medicines Association term of 'falsified medicine' is used in relation with the Falsified Medicines Directive (FMD) while the FDA term of counterfeit medicine is used in relation with the Drug Supply Chain Security Act (DSCSA).

Medicine counterfeiting is a global problem which requires international action.[5] According to the pharmaceutical security institute (PSI), there has been a 34% increase in international drug counterfeiting, illegal diversion and theft in the last year, and a 51% increase since 2011.[6] Countries around the world have developed a number of methods to detect counterfeit drugs which range from laboratory-based and portable Raman and Quadrupole spectroscopy in developed countries[7–10] to SMS messaging in low-income and developing countries.[11]

This study assesses serialisation and verification technology, the most rapidly emerging method for detecting counterfeit medicines. This technological approach has been adopted in the European Union (EU) under the FMD[12–14] and in the USA under the DSCSA.[15] Authentication relates to the verification of a medicines authenticity and subsequent decommissioning using database cross checking of the medicines unique 12-digit serial code, at the point of supply to the patient. Verification refers to the scanning of a medicinal product to identify its status in terms of falsification, expiry or recalled status without decommissioning. The EU FMD prescribes the systematic serialisation at manufacture and authentication at the point of supply to the patient while the DSCSA requires the verification of medicinal products at each change of ownership without verification or decommissioning at the point of supply to the patient.

The European Medicines Verification Organisation (EMVO) has described the core principles of authentication technology and has appointed three blueprint authentication technology providers for Europe. These providers are Aegate Holdings Limited, Arvato Systems GmbH and Solidsoft Reply. Each of these providers is expected to meet the minimum requirements as set by the EMVO.[16] As only one authentication technology, provided by Aegate Holdings Limited, was operational at the time of research, it was chosen to be integrated into a live secondary care hospital environment as part of a service evaluation study. On the scanning and subsequent detection of a recalled, expired or potentially counterfeit medicine, this technology presents the user with red or amber pop up messages which identifies a product as requiring quarantine. Medicines authenticated which do not require quarantine do not generate a pop up but instead generate a 'non-action' symbol. On integration, staff were trained via presentation and demonstration and also issued with an authentication protocol which detailed step by step instructions on how to authenticate medicines.

The key results from the service evaluation study related to authentication rates and detection rates of the technology. Although the technical detection rate of the technology was 100% and the response rate of the technology was <300 ms (as mandated by the FMD),[13 14] not all medicines requiring authentication were authenticated and of those authenticated not all of those that generated a warning pop up were quarantined.[17] Following the service evaluations study, this Delphi method study was conducted to identify the reasons for poor authentication and detection rates and identify improvements.

Despite the 2019 legislative technology compliance deadline for pharmacies, dispensing general practitioner practices and hospitals across Europe, there is little qualitative evidence to support this international technological approach to counterfeit drug detection in practice. If implemented incorrectly this international change has the potential to cause considerable upset for healthcare providers. This study aims to qualitatively evaluate and inform the optimisation of medicine authentication technology in secondary care.

A group of 10 secondary healthcare professionals with experience in medicines authentication were surveyed following a service evaluation study conducted in 2015, using the Delphi method approach. The Delphi method approach, originally used as a systematic forecasting tool,[18 19] has been increasingly used to gain consensus expert opinion and aid decision-making in a variety of research areas. Delphi methodology provides clearer outcomes and recommendations than traditional surveys often produce, achieved by collecting responses and summarising responses until a consensus is achieved.

As part of the FMD, policymakers and key opinion leaders across Europe must come together to form National Medicines Verification Organisations (NMVO). Each NMVO will make decisions regarding authentication of technology providers which may relate to functionality, speed of data response, usability and technology limitations. The study aims to identify the positives of the incoming technology and educate decision makers as to the potentially useful technological changes that could be adopted or mandated as part of the incumbent medicines authentication technology.

## METHODS

Twelve participants from a UK hospital pharmacy department with experience of using medicines authentication technology as part of a service evaluation project and satisfied the study inclusion and exclusion criteria (box 1) were invited to take part in a Delphi method study. An implied consent model was used, with information regarding the nature of the study contained in the invitation email[18–22] and repeated on the first page of the questionnaire. This information described the study and its voluntary nature. From a total population size of 12, 11 invitations were accepted. One staff

**Box 1** Inclusion and exclusion criteria for study participants

*Inclusion criteria:*
▶ Staff accredited in the professional checking process at the test site*.
▶ Staff with multiple experiences of using the authentication technology.
▶ Staff willing to complete multiple surveys.
▶ Staff who have attended basic training as part of the service evaluation project on medicines authentication.

*Exclusion criteria:*
▶ Staff who have used the system once or not at all.
▶ Staff who have not passed the trust checking accreditation test.

*Staff were a mixture of General Pharmaceutical Council-registered pharmacists and qualified accuracy checking technicians (nationally recognised qualification).

member was unavailable due to sickness and one staff member was initially invited inappropriately, as they did not meet the inclusion criteria. A total of 100% (n=10) of available eligible participants responded (95% CI) (2% margin of error).

Prospective participants were asked to complete an electronic questionnaire[18 23] with an estimated completion time of 15 min or less. Participants received a total of two questionnaires[21] (ie, one for each round of iteration of the questionnaire). The initial invite was followed by a number of reminders at ~8-day intervals until completion. The question and response format included three different categories of questions (Box 2), a 7-point rating scale, a Likert scale and a descriptive or open-ended response format. Likert scales were used to prioritise suggestions. In some cases, the staff identified four suggestions and in some cases they identified five suggestions, therefore two Likert scales were employed.

## Consensus
The 7-point rating scale consensus was achieved when 70% of respondents selected either of two adjacent

**Box 2** Question and response types

▶ 7-Point Rating Scale (r): This question format rates performance from 1 to 7, with '1' indicating a negative response and '7' indicating the most positive .The median response was also gathered for further consensus evaluation.

▶ Likert Scale response format[9] (l) : This question style requires respondents to prioritise suggested improvements in terms of importance from '1' to '4' or '1' to '5' with '1' indicating most important and '5' indicating least important (median).

▶ Descriptive/ open-ended response format (d): This response format does not require prioritisation or rating; rather the analysis of responses generates a consensus descriptive answer for the group (median).

answers on a 7-point rating scale[24] and the median score fell within either of the two consensus answers[25] (table 1). In terms of the 5-point Likert scale, response format[24] consensus was achieved when 80% of the respondents selected one of three adjacent answers in either direction that is, whether participants classed the suggestion as 'important' (1–3) or 'not important' (3–5). On the 4-point Likert scale, the same rules were applied, however, 'important' was classified as 1–2 and 'not important' was classified as 2–4 with consensus being achieved when 70% of respondents selected one of two answers in either direction (table 1). In the case of a 5-point scale, the median must also have been below 2.5, and in consideration of a 4-point scale the median must also have been below 2 to be considered 'important'[19] (table 1).

In terms of the 5-point descriptive/open-ended response format scale, consensus was assessed depending on the theme of the response, for example, if 80% of respondents selected one of three descriptive adjacent answers of the same sentiment (positive or negative) on a 5-point scale, the median score fell within the consensus category.

The planning of this study was based largely on the highly cited Hsu and Sandford paper from 2007 which describes evidence for successful Delphi studies such as number of survey rounds, consensus figures and thematic analysis.[19] Delphi method consensus varied across studies and opinions differ on a suitable percentage measure. This study compliments those conducted by Green in 1982[24] and Ulschak in 1983[25] (in Hsu and Sandford in 2007)[19] which promote a consensus of 70 and 80%, respectively. The default consensus in this study was 70% with 80% used where possible.

SD was chosen as a method to measure central tendencies with an arbitrary <1.0 SD used to supplement the consensus. This quantified the spread of responses across the entire group.

## Summary of survey rounds
Round one involved three demographic questions, followed by a selection of closed questions relating to rating scales and performance and open questions requiring suggestions for a number of improvements, followed by descriptive questions to evaluate the quality of the survey. A selection of questions was built on feedback from users, during a study which involved the use of authentication technology for an 8-week period.[17]

The administration of the second questionnaire was similar to round one; closed questions that achieved consensus were removed. Non-consensus questions were resubmitted to the audience in a rating scale style with further explanation.

Answers to the open-ended questions in round one were thematically categorised and summarised to remove duplicate suggestions. In round two, the experts were asked to answer further questions based on the most frequently occurring themes, which included

**Table 1** Summary of consensus

| Question type | Consensus (%) | Median response | Consensus description |
|---|---|---|---|
| 7-Point Rating Scale | 70% agreement | Must fall within the consensus category | One of two adjacent answers |
| Likert Scale 5 point | 80% agreement | <2.5 (important) | One of three adjacent answers in either direction |
| Likert Scale 4 point | 70% agreement | <2 (important) | One of three adjacent answers in either direction |
| Descriptive/open | 80% agreement | Must fall within the consensus category | One of three adjacent answers of the same sentiment (positive or negative) |

Likert scales or descriptive style questions formed as a direct result of participant suggestions during round one of the study or during the service evaluation study.[17] The total number of suggestions per question varied between four and five suggestions which directly affected the number of options available in round two.

A valid consensus result was considered as 'achieved' when consensus had been met and the median scores also fell within the consensus group (table 1). Suggestions that fell within these parameters and had a SD of <1.0 were considered as the most relevant improvements for the authentication technology. There was no follow-up with participants after study completion.

## RESULTS
A selection of questions which reached immediate consensus is described in tables 2 and 3. Answers in green boxes identify the results which reached consensus with a narrow SD (<1.0). Answers in amber identify a consensus response with a larger SD (>1.0).

The most important suggestions in this study are those which established consensus among the group (table 1) with a narrow SD. Improvements included reviewing the colour and information in warning pop up screens to ensure they were not mistaken for the 'already dispensed here' pop up (2.0) (SD 0.87), encouraging the dispenser/checker to act on the warnings (2.0) (SD 1.0) (Education), including an audible alert to accompany the pop up warning box (1.0) (SD 0.9) (Technology) and making it mandatory to complete an 'action taken' documentation process to improve the quarantine process for potentially counterfeit, expired or recalled medicines. (Technology) (2.0) (SD 0.94).

## Discussions and recommendations
The serialisation and authentication of medicines has been proposed in response to international regulation. Considering that serialisation and verification with or without authentication will affect EU hospital pharmacies and is likely to affect US hospitals wishing to wholesale supply, it is important to gauge its current appropriateness and identify improvements in this valuable patient safety technology.

There were no concerns regarding the speed and usability of authentication technology raised during this study. There was limited impact on the daily activity of the staff and was classed as 'not disruptive'. Further qualitative research would be useful in understanding

**Table 2** A summary of 7-Point Rating Scale results

| No | Question | Result | SD |
|---|---|---|---|
| **Round 1** | | | |
| 4 | Based on your experience of the Medicines Authentication System (MAS), how would you rate its general speed on a scale of 1 to 7? (1 represents very slow and 7 represents very fast) | 6/7 | 0.75 |
| 5 | Based on your experience of the MAS, how would you describe its usability on a scale of 1 to 7? (1 represents very difficult and 7 represents very easy) | 6.5/7 | 0.87 |
| 6 | There were some system errors reported by the MAS users throughout the pilot. On a scale of 1 to 7, how often did you experience these types of errors? (1 represents never and 7 represents very often) (These errors may have included issues with reading the two dimensional barcode, duplication of the scan on screen or warnings such as 'The system has no resources', 'item can is invalid please scan product again' or 'test product not found') | 1/7 | 0.67 |
| **Round 2** | | | |
| 7 | Question 4: How would you rate the impact of the MAS on the service you provide on a scale of 1–7 (where 1 represents very disruptive and 7 represents very helpful)? | 4/7 (Not disruptive) | 0.94 |

**Table 3** A summary of 4 and 5-Point Likert-like scale results

| No | Question | Result | SD |
|---|---|---|---|
| **Round 1** | | | |
| 7 | The following is a list of reported, proposed improvements. Please rank them in order of importance (1–5). | | |
| 7 (ii) | Change the medicine scanning list on screen to ensure the last scanned item appears on the top of the list. | 1 (Important) | 1.25 |
| 7 (iv) | Review the pop up screens as the Red 'Warning' screens could be mistaken for the common 'Already dispensed here' screen. | 2 (Important) | 0.87 |
| 7 (v) | Incorporate 'important information' pop ups into the authentication system. | 2.5 (Important) | 1.25 |
| **Round 2** | | | |
| 5 | During round one of this survey, further suggestions were made to improve the Medicines Authentication System (MAS) or the pilot. Please rank the suggested changes below in order of importance (1 being most important and 5 being least important). | | |
| (ii) | Sounds could also be enabled to ensure warnings/information are noticed (MAS). | 2.5 (Important) | 1.2 |
| 6 | During round one of this survey there were a variety of suggestions made to increase the rate of authentication (scanning). Valid suggestions were subdivided into three categories 1. process change 2.technology change and 3.education. In terms of process change, please rank these suggestions in order of importance (with 1 being the most important and 5 being the least important). | | |
| (i) | Make the symbol indicating an item that needs to be scanned larger/more visible (process). | 2.5 (Important) | 0.92 |
| 7 | During round one of this survey, there were a variety of suggestions made to increase the rate of authentication (scanning). Valid suggestions were subdivided into three categories 1. process change 2. education and 3. technology change. In terms of education and technology change, please rank these suggestions in order of importance (with 1 being the most important and 5 being the least important) | | |
| (iv) | A system change that knows how many items have been booked in and prescription is not able to be tracked out as verified until all medications have been authenticated (technology change) | 2.5 (Important) | 1.2 |
| 8 | During round one we explained that there have been occasions where products have been handed out despite showing a pop up warning box. We asked you to list three suggestions of how this occurrence might be reduced. Valid suggestions were subdivided into two categories 1. education and 2. technology change. In terms of education, please rank these suggestions in order of importance (with 1 being the most important and 4 being the least important) | | |
| (i) | Encourage the dispenser/checker to take action on the warnings (education) | 2.0 (Important) | 1.0 |
| 9 | During round one we explained that there have been occasions where products have been handed out despite showing a pop up warning box. We asked you to list three suggestions of how this occurrence might be reduced. Valid suggestions were subdivided into two categories 1. education and 2. technology change. In terms of technology change, please rank these suggestions in order of importance (with 1 being the most important and 4 being the least important) | | |
| (ii) | An audible alert to accompany the pop up warning box (technology) | 1.00 (Important) | 0.9 |
| (iii) | Making it mandatory to complete an 'action taken' documentation process so that staff scanning are prompted to think about what the red warning means and be accountable for it (technology) | 2.00 (Important) | 0.94 |

why the participants considered this technology as not disruptive.

The Delphi study identified a number of suggestions to improve authentication and detection rates including the importance of making clear differentiation between the various warning messages to avoid misinterpretation. Participants identified concern with the similarity of warning messaging which may have had an impact on the decision to quarantine and may have contributed to the suboptimal detection rates seen in previous publications relating to the service evaluation study.[17] This study recommends that authentication technology providers test their technology in a closed loop and real-life environment prior to national implementation to ensure that operators respond appropriately to the messaging being displayed.

As with all changes to practice, adequate education and training is required. There was a basic presentation and protocol approach used in the service evaluation study[17] which may have contributed to inadequate

authentication and detection rates due to lack of adequate support or perhaps due to the sheer volume of procedures and protocols that staff are required to adhere to. A practice change of this size may benefit from a structured training and revalidation package. If we refer to previous large information technology projects such as the implementation of the electronic patient record (EPR), we can see a much more complicated electronic system with a structured training package which largely includes presentations, demonstrations, workshops, drop-in sessions, protocols and guidance. A varied, informed and interactive approach, as used in the EPR project, is required to build operator background knowledge and to instil the clinical and legal importance of authenticating medicines as well as the impact of authentication on medicine detection rates.

The recent quantitative service evaluation study identified a disparity between medicines identified as requiring quarantine and those actually quarantined.[17] This qualitative study demonstrated a consensus that a pop up warning message alone is not an adequate prompt in the heterogeneous hospital environment.[10] The operators in this study are in agreement that a noise to indicate a medicine as requiring quarantine is required. This would remind the operator to act and would also bring attention to the entire team that a medicine requiring quarantine had been identified, generating peer pressure to act on the warning.

Participants emphasised the importance of an 'action taken' documentation process. Operators relied on recording the medicine for quarantine on a proforma located beside the terminal; their view was that this could be improved with the inclusion of an 'action taken' function incorporated into the authentication technology software. In a systematic review study by Shojania et al, the action taken alert has been identified as potentially more effective than a 'non-action' alert[26] demonstrated in a small group. This 'action taken' approach may facilitate an improved detection rate and may also support a reporting system which would benefit managerial monitoring of falsified, counterfeit, recalled or expired medicines within a department. This would also allow staff responsible for product quarantine to tally medicines physically quarantined with medicines identified by the technology as requiring quarantine.

Information technology systems such as electronic prescribing or in this case medicines authentication are relatively new approaches to optimise healthcare information. Shojania et al.[26] demonstrate that evidence to support computer alerts is currently limited. In relation to electronic prescribing, Shojania concludes by stating 'Further research must identify design features and contextual factors consistently associated with larger improvements in provider behaviour if computer reminders are to succeed on more than a trial and error basis'. Further research is also required in relation to medicines authentication technology to identify the approaches which facilitate operational compliance. Hospitals in the NHS vary slightly depending on the services they provide, which makes context an important factor in technology success.[27] Following on from the remarks made by Shojania et al, further research is required to understand how contextual factors can facilitate successful technological projects in the NHS. One such contextual factor may include incentives to authenticate medicines. The use of reimbursement codes within the two dimensional data matrix codes is hypothesised to help the authentication rate of medicines, a practice seen in the Belgian community pharmacy setting. There will be a legal mandate to authenticate medicines; however, incentives such as reimbursement on authentication may prove to augment authentication and detection rates and legal compliance.

The results from this study should inform key opinion leaders, policymakers and technology manufacturers as to the limitations of medicines authentication technology and potential authentication technology improvements. Considering the limited evidence to support medicines authentication, the outcomes of this study should also service decision makers in their discussions surrounding the selection of medicines authentication technology providers.

**Author affiliations**
[1]Institute of Science and Technology in Medicine, Keele University, Keele, UK
[2]Said Business School, University of Oxford, Oxford, UK
[3]Pharmacy Department, Oxford University Hospitals NHS Trust, Oxford, UK
[4]Medicines Optimisation Clinical Network, Oxford Academic Health Science Network (AHSN), Oxford, UK
[5]Department of Paediatrics, University of Oxford, Oxford, UK
[6]The Oxford–UCL Centre for the Advancement of Sustainable Medical Innovation (CASMI), University of Oxford, Oxford, UK
[7]Centre for Behavioural Medicine, UCL School of Pharmacy, University College London, London, UK
[8]Harvard Stem Cell Institute, Cambridge, Massachusetts, USA
[9]UCSF-Stanford Centre of Excellence in Regulatory Science and Innovation (CERSI), San Francisco, California, USA

**Collaborators** Bhulesh Vadher, Chief Pharmacist, Oxford University Hospitals, NHS Foundation Trust.

**Contributors** BN and DB were responsible for study conception; BN, DB, LR, SD and SC were responsible for planning; BN was responsible for data collection and scripting; BN and LR were responsible for data analysis; BN, DB, LR, SD and SC were responsible for the reviewing of this manuscript.

**Funding** This study was funded by Keele University, Oxford University and Aegate limited via an unrestricted educational grant.

**Disclaimer** The content outlined herein represents the individual opinions of the authors and may not necessarily represent the viewpoints of their employers.

**Competing interests** DB is an employee and/or stockholder in Aegate (Melbourn, UK) that is a provider of medicines authentication services. DB is also a stockholder in Translation Ventures (Charlbury, Oxfordshire, UK) and IP asset ventures. DB is subject to the CFA Institute's Codes, Standards and Guidelines, and as such, this author must stress that this piece is provided for academic interest only and must not be construed in any way as an investment recommendation. BN is an advisory board member for IDIS, part of the Clinigen group. BN is currently not, but has previously been a consultant of Aegate Ltd. Additionally, at time of publication, DB and the organisations with which he is affiliated may or may not have agreed and/or pending funding commitments from the organisations named herein.

**Provenance and peer review** Not commissioned; externally peer reviewed.

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
