## [Reviewer comments · BMJ Open]

ARTICLE DETAILS

TITLE (PROVISIONAL)	Medicine Authentication Technology as a Counterfeit Medicine Detection Tool : A Delphi Method Study to Establish Expert Opinion on Manual Medicine Authentication Technology in Secondary Care
AUTHORS	Naughton, Bernard; Roberts, Lindsey; Dopson, Sue; Brindley, David; Chapman, Stephen

VERSION 1 - REVIEW

REVIEWER	Tim Mackey UC San Diego - School of Medicine
REVIEW RETURNED	08-Sep-2016

GENERAL COMMENTS	General comments ===== Thank you for the opportunity to review this manuscript that conducts a DELPHI consensus study to examine medicine authentication systems in the context of counterfeit medicines regulation. ABSTRACT: -Authors should refer to Drug Supply Chain Security Act (DSCSA) not H.R. bill US Drug Quality and Security Act. Generally I didn't think the abstract reflected the study aims or the results of the study. The abstract was much more general to the problem of SSFFC but should instead focus on what the MAS is, Delphi study results to improve the system, and implications of findings. INTRODUCTION -first para: Authors should start with the proper definition of the problem and also a brief description of the definitions for SSFFC, this should later include discussion about how these definitions fit into the language of the FMD and DSCSA. Authors should also include Attaran et al. 2012 BMJ piece in first sentence reference. Authors should also specifically name source of data reporting increase (i.e. PSI). Last 3 sentences the implementation requirements for FMD need to be better described to the reader and aligned with the study aim. Is the point to measure the appropriate technology to ensure authentication within implementation of FMD (now that it is already enacted) or to establish best practices? This is partially explained in the last sentence of the last paragraph of this section, but could be better clarified. If it is an evaluation of only one MAS, then the implications are relatively limited. METHODS -first para: in relation to the inclusion and exclusion criteria it would be good to list the types of accreditation and types of authentication technology that participants use. This would provide better context. Authors should also explicitly state that the study received IRB approval. -consensus: since table 3.0 summarizes the consensus paragraph
---

wouldn't this section be better used to explain why these thresholds are set and how they support consensus on the topic based on prior studies/literature?

RESULTS

-first para: If results and questions from this study are limited to the "Medicines Authentication Systems (MAS)" then explaining to the readership exactly what this constitutes needs to be done much earlier even if this was done in a separately published evaluation or an under review/submitted paper. What type of authentication system is this? Is it only one type of technology and one protocol for all types of medicines, or is it made up of multiple devices (barcode reader, mass spec, etc?) This really needs to be explained earlier, and the implications of your results limited to evaluation of this system. If there are multiple authentication systems used by different respondents, then that could influence study results. If it is one single system, that is not currently made clear in the introduction or methods section. Also, given the COI included in this manuscript, alternative authentication systems and what services they provide should be discussed in an objective manner and as alternatives to the MAS.

-are there any questions that specifically relate to how respondents were trained on the system and what type of training they need? I did not see this.

-p.13, as written in the results section, there is no context for the results presented (such as the warning pop up screens, etc) so the readership has no idea why the results are important or significant to improving the MAS technology. It is even farther away from adding to understanding of how this system and its technology could help in implementation of the FMD. This needs better explanation or the results won't mean anything to the reader.

RESULTS

-first para, why does the specific MAS examined impact the entire US and EU market. Has this particular MAS been adopted as a standard for authentication technology, I'm not aware that it has, especially in the US where the DSCSA is still just under a decade away of full implementation. The results of the study do not support this statement.

-there is no formal limitations section, that would need to be added.

Overall, this research paper has many flaws as described above and I cannot recommend for publication without substantial revision. The piece looks like part of a larger study/technology assessment of the MAS, but without the mentioned service evaluation (which has also been submitted to BMJ Open) there is no way to evaluate this piece. Perhaps authors should think about combining the two pieces so that it covers both aspects as BMJ Open allows a high word count. Right now there is no context regarding the MAS or how it fits into the overall FMD requirements and implementation schedule. The link to DSCSA is even farther off. Right now as presented, I'm not sure a general readership would understand what the study is evaluating and why it is important. I also think the implications of findings are limited to a technology assessment of the MAS and cannot be extrapolated to larger policy implementation issues. I like the incentives discussion at the end of pg.15, but right now it doesn't link with the results. I think the piece would be better suited if combined with the other evaluation or sent to a journal that specifically addresses technology assessments.

REVIEWER	Mustapha Hajjou U.S. Pharmacopeial Convention USA
REVIEW RETURNED	14-Sep-2016

GENERAL COMMENTS	Page 1; line 3: It is recommended to use the word Falsified instead of Counterfeit. This should be applied throughout the paper. The abstract is not well balanced. There is only one sentence on the results of the study. The statement in the first sentence lacks accuracy (see comment for page 3). Page 2; line 13-15: The statement is a repeat from the introduction (see comment below). It should be revised. Page 3; line 16-18: The statement about % increase of drug counterfeiting seems erroneous (reference (2)). The numbers reported on the PSI website are for incidents of pharmaceutical counterfeit, illegal diversion, and theft. The 51% increase reported is for pharmaceutical crime not just drug counterfeiting. This part of the introduction should be re-written or deleted. Page 3; line 43: Please spell out GP Page 7; line 49: How the reference will appear in the final paper if this manuscript is accepted for publication? Page 15; line 3-7: The authors made assertion based on a qualitative study submitted for publication. Since that manuscript has not been accepted for publication yet, I am not sure the assertion and the reference should be made. Similar to the question Page 15, how the reference will appear in the paper is the present manuscript is accepted for publication. Is it assumed that the manuscript referenced will be accepted? It would probably make the present manuscript stronger if the two manuscripts were combined.
--

VERSION 1 – AUTHOR RESPONSE

Reviewer: 1

Reviewer Name: Tim Mackey

Institution and Country: UC San Diego - School of Medicine Please state any competing interests or state 'None declared': none declared

Please leave your comments for the authors below General comments ===== Thank you for the opportunity to review this manuscript that conducts a DELPHI consensus study to examine medicine authentication systems in the context of counterfeit medicines regulation.

ABSTRACT:

-Authors should refer to Drug Supply Chain Security Act (DSCSA) not H.R. bill US Drug Quality and Security Act. Generally I didn't think the abstract reflected the study aims or the results of the study. The abstract was much more general to the problem of SSFFC but should instead focus on what the MAS is, Delphi study results to improve the system, and implications of findings.

Agreed, the abstract has been restructured according to BMJ open guidelines and DSCSA is referred to in place of DQSA this publication.

INTRODUCTION

-first para: Authors should start with the proper definition of the problem and also a brief description of the definitions for SSFFC, this should later include discussion about how these definitions fit into the language of the FMD and DSCSA.

Authors should also include Attaran et al. 2012 BMJ piece in first sentence reference (agreed and completed). Authors should also specifically name source of data reporting increase (i.e. PSI) agreed and completed). Last 3 sentences the implementation requirements for FMD need to be better described to the reader and aligned with the study aim. Agreed and completed. Is the point to measure the appropriate technology to ensure authentication within implementation of FMD (now that

it is already enacted) or to establish best practices? The point is to identify positives and negatives associated with using medicines authentication technology from an end user point of view. This has been explained more clearly as per the recommendation above.

This is partially explained in the last sentence of the last paragraph of this section, but could be better clarified. If it is an evaluation of only one MAS, then the implications are relatively limited. At the time of conducting this study this was the only commercially available authentication technology available for evaluation. Considering that the EMVO (European medicines verification organisation) have mandated that all companies creating these technologies comply to a blueprint service. Therefore there will be only minor difference between technologies offered by different companies. It was considered necessary to evaluate the available technology to inform policy makers of the pitfalls before deciding on a national provider.

METHODS

-first para: in relation to the inclusion and exclusion criteria it would be good to list the types of accreditation and types of authentication technology that participants use. This would provide better context. Authors should also explicitly state that the study received IRB approval.

Agreed, the accreditation and types of technology used have now been listed. This study did not require ethics approval as it was classed as a service implementation project according to the UK health research authority guidelines and as agreed by the research steering committee. The reasons for this include the lack of patient participants and the fact that this technology was not altered for this study i.e the same system as used in other European community pharmacies. For more information regarding the classification of a service evaluation study please see the link below.
<http://www.hra.nhs.uk/documents/2016/06/defining-research.pdf>

-consensus: since table 3.0 summarizes the consensus paragraph wouldn't this section be better used to explain why these thresholds are set and how they support consensus on the topic based on prior studies/literature?

I agree that we should add to the consensus paragraph a little about the rationale for choosing the thresholds, please see amended manuscript. We included the table to summarise the text and to ensure that those unfamiliar with Delphi studies would be able to understand the study structure. We believe that the table alone may not suffice.

RESULTS

-first para: If results and questions from this study are limited to the "Medicines Authentication Systems (MAS)" then explaining to the readership exactly what this constitutes needs to be done much earlier even if this was done in a separately published evaluation or an under review/submitted paper. What type of authentication system is this? Is it only one type of technology and one protocol for all types of medicines, or is it made up of multiple devices (barcode reader, mass spec, etc?) This really needs to be explained earlier, and the implications of your results limited to evaluation of this system. If there are multiple authentication systems used by different respondents, then that could influence study results. If it is one single system, that is not currently made clear in the introduction or methods section. Also, given the COI included in this manuscript, alternative authentication systems and what services they provide should be discussed in an objective manner and as alternatives to the MAS.

We agree with this opinion. We have now introduced a paragraph in the introduction identifying clearly the technology being studied, identifying that we are testing a single example of this database checking technology.

All three providers have now been listed in the introduction and although we are testing one provider of authentication technology its core functionality is mimicked by other providers as mandated by the EMVO.

-are there any questions that specifically relate to how respondents were trained on the system and what type of training they need? I did not see this.

We have now explained that the participants were trained with a presentation, provided with a protocol on how to authenticate and also given a demonstration.

-p.13, as written in the results section, there is no context for the results presented (such as the warning pop up screens, etc) so the readership has no idea why the results are important or significant to improving the MAS technology. It is even farther away from adding to understanding of

how this system and its technology could help in implementation of the FMD. This needs better explanation or the results won't mean anything to the reader.

Further information has now been added to the introduction which will add context to the results section.

RESULTS

-first para, why does the specific MAS examined impact the entire US and EU market. Has this particular MAS been adopted as a standard for authentication technology, I'm not aware that it has, especially in the US where the DSCSA is still just under a decade away of full implementation. The results of the study do not support this statement.

Medicines authentication technology must adhere to the EMVO standard, the 'Blueprint' as it is referred to. As this is one of the blue print providers it demonstrates the core function of an authentication technology. Therefore, the problems associated with this technology in terms of database communication and staff useability will relate to other technologies available in Europe. It is important for the US to be aware that although verification and authentication will not be conducted at the point of supply to the patient it may be conducted in hospitals should that hospital wish to wholesale supply, therefore the verification issues demonstrated here may appear also in the US hospital sector. This was not been made clear and has now been explained further in the manuscript.

-there is no formal limitations section, that would need to be added.

The limitations section is contained under the abstract and the limitations that this reviewer has mentioned above have now been included.

Overall, this research paper has many flaws as described above and I cannot recommend for publication without substantial revision. The piece looks like part of a larger study/technology assessment of the MAS, but without the mentioned service evaluation (which has also been submitted to BMJ Open) there is no way to evaluate this piece.

This is part of a larger study; however, the service evaluation study is an already very large publication. We have now substantially revised this manuscript and have described the technology and previous study better in the introduction of this paper, which contextualises the results.

Perhaps authors should think about combining the two pieces so that it covers both aspects as BMJ Open allows a high word count. Right now there is no context regarding the MAS or how it fits into the overall FMD requirements and implementation schedule.

As stated above the service evaluation study which was researched alongside this piece is already too large to be published alongside this piece. The context has now been added which sets the scene for the results.

The link to DSCSA is even farther off. Right now as presented, I'm not sure a general readership would understand what the study is evaluating and why it is important. I also think the implications of findings are limited to a technology assessment of the MAS and cannot be extrapolated to larger policy implementation issues. The DSCSA link was tangential, we have now explained the exact stakeholder which these results may benefit i.e the hospital wholesaler. The service evaluation contained the context for this study, we now appreciate that information regarding the previous study needs to be part of this manuscript and have therefore added it.

I like the incentives discussion at the end of pg.15, but right now it doesn't link with the results.

We have inserted some text which highlights the work of Shojania et al which demonstrates some academic evidence to support one of the suggestions made by the staff. Shojania also explains that there is limited evidence to support certain pop ups/ alerts and that further research is required to understand which alert types and contextual factors bring about consistent behaviour changes. This brings together the study results and the discussion surrounding the importance of contextual factors which links in incentives.

I think the piece would be better suited if combined with the other evaluation or sent to a journal that specifically addresses technology assessments.

We have added the context which is contained in the other evaluation, which we believe provides the background necessary to appreciate these results.

Reviewer: 2

Reviewer Name: Mustapha Hajjou

Institution and Country: U.S. Pharmacopeial Convention, USA Please state any competing interests or state 'None declared': I do not have any competing interest

Please leave your comments for the authors below

Page 1; line 3: It is recommended to use the word Falsified instead of Counterfeit. This should be applied throughout the paper.

The terms falsified, counterfeit and SFFC have been described in the introduction as requested by the initial reviewer.

The abstract is not well balanced. There is only one sentence on the results of the study. The statement in the first sentence lacks accuracy (see comment for page 3).

The Abstract has now been restructured as per BMJ OPEN guidelines

Page 2; line 13-15: The statement is a repeat from the introduction (see comment below). It should be revised. This sentence has been removed from the abstract and has been re-worded in the introduction to ensure the statement is accurate.

Page 3; line 16-18: The statement about % increase of drug counterfeiting seems erroneous (reference (2)). The numbers reported on the PSI website are for incidents of pharmaceutical counterfeit, illegal diversion, and theft. The 51% increase reported is for pharmaceutical crime not just drug counterfeiting. This part of the introduction should be re-written or deleted.

This sentence has now been re-written to state 'According to the pharmaceutical security institute (PSI) there has been a 34% increase in international drug counterfeiting, illegal diversion and theft' This now fits in better as it is preceded by a definition of counterfeit, falsified and SFFC medicine.

Page 3; line 43: Please spell out GP

Completed

Page 7; line 49: How the reference will appear in the final paper if this manuscript is accepted for publication?

This paper is likely to be accepted we have therefore referenced it as it is likely to appear in the journal.

Page 15; line 3-7: The authors made assertion based on a qualitative study submitted for publication. Since that manuscript has not been accepted for publication yet, I am not sure the assertion and the reference should be made.

This paper is likely to be accepted we have therefore referenced it as it is likely to appear in the journal.

Similar to the question Page 15, how the reference will appear in the paper is the present manuscript is accepted for publication. Is it assumed that the manuscript referenced will be accepted? This paper is likely to be accepted we have therefore referenced it as it is likely to appear in the journal.

It would probably make the present manuscript stronger if the two manuscripts were combined

This point has also been made by the previous reviewer. This would not be possible however as the first quantitative study is already quite large. We appreciate your point of view and have therefore given a brief of what occurred in the quantitative study which is pending acceptance, this places this study (Delphi study) into better context which makes the results more meaningful.

VERSION 2 – REVIEW

REVIEWER	Mustapha Hajjou United States Pharmacopeial Convention
REVIEW RETURNED	31-Jan-2017

GENERAL COMMENTS	Introduction: Line 11-15: WHO now uses the term SF (substandard and falsified)
------------------	---

	instead of FFSSC. Please update
--	---------------------------------